# Renal Oxygen Demand and Nephron Function: Is Glucose a Friend or Foe?

**DOI:** 10.3390/ijms24129957

**Published:** 2023-06-09

**Authors:** Edoardo Gronda, Alberto Palazzuoli, Massimo Iacoviello, Manuela Benevenuto, Domenico Gabrielli, Arduino Arduini

**Affiliations:** 1Medicine and Medicine Sub-Specialties Department, Cardio Renal Program, U.O.C. Nephrology, Dialysis and Adult Renal Transplant Program, IRCCS Ca’ Granda Foundation, Ospedale Maggiore Policlinico, 20122 Milano, Italy; 2Cardiovascular Diseases Unit, Cardio Thoracic and Vascular Department, S. Maria alle Scotte Hospital University of Siena, 53100 Siena, Italy; palazzuoli2@unisi.it; 3Department of Medical and Surgical Sciences, University of Foggia, 71100 Foggia, Italy; 4Unità Operativa Complessa Cardiologia-UTIC-Emodinamica, PO Giuseppe Mazzini, 64100 Teramo, Italy; manuelabenvenuto82@gmail.com; 5Unità Operativa Complessa Cardiologia-UTIC, Azienda Ospedaliera San Camillo Forlanini, 00152 Rome, Italy; domgab61@gmail.com; 6R&D Department, CoreQuest Sagl, 6900 Lugano, Switzerland; a.arduini@corequest.ch

**Keywords:** glucose transport, SGLT2 inhibitor, diabetic nephropathy, heart failure, chronic kidney disease

## Abstract

The kidneys and heart work together to balance the body’s circulation, and although their physiology is based on strict inter dependence, their performance fulfills different aims. While the heart can rapidly increase its own oxygen consumption to comply with the wide changes in metabolic demand linked to body function, the kidneys physiology are primarily designed to maintain a stable metabolic rate and have a limited capacity to cope with any steep increase in renal metabolism. In the kidneys, glomerular population filters a large amount of blood and the tubular system has been programmed to reabsorb 99% of filtrate by reabsorbing sodium together with other filtered substances, including all glucose molecules. Glucose reabsorption involves the sodium–glucose cotransporters SGLT2 and SGLT1 on the apical membrane in the proximal tubular section; it also enhances bicarbonate formation so as to preserve the acid–base balance. The complex work of reabsorption in the kidney is the main factor in renal oxygen consumption; analysis of the renal glucose transport in disease states provides a better understanding of the renal physiology changes that occur when clinical conditions alter the neurohormonal response leading to an increase in glomerular filtration pressure. In this circumstance, glomerular hyperfiltration occurs, imposing a higher metabolic demand on kidney physiology and causing progressive renal impairment. Albumin urination is the warning signal of renal engagement over exertion and most frequently heralds heart failure development, regardless of disease etiology. The review analyzes the mechanisms linked to renal oxygen consumption, focusing on sodium–glucose management.

## 1. Introduction

Sodium–glucose cotransporter 2 inhibitors (SGLT2i) are a relatively novel class of molecules named gliflozines that were originally designed to control glycemia, but over and above their effect on the glycemic metabolism, they unexpectedly improved the management of clinical conditions involving the function of the heart and kidneys. Importantly, the benefits of SGLT2i go beyond guideline-directed medical therapy, in that the drug’s metabolic effects on the kidneys and heart ameliorate the entire cardio–circulatory outcome.

Since it has been researched, SGLT2i has been proven to decrease hemoglobin A1c by ≈0.7% to 1.0% and to induce weight loss through glycosuria; through concomitant natriuresis and osmotic diuresis, it succeeds in contracting the plasma volume by ≈7%, contributing ≈5/2–mm Hg antihypertensive effects; lastly, it has also been proven to reduce albuminuria by 30–40% through the intrarenal hemodynamic effect [1]. Notwithstanding the apparently limited range of such benefits, a wealth of investigations performed using SGLT2i have revealed that this class of drug can affect the cardiovascular outcome in diabetics with [2] and without cardiovascular damage [3], diabetics and non-diabetics with advanced CKD with and without renal albuminuric insult [4,5,6], and diabetic and non-diabetic heart failure (HF) patients especially via the ventricular ejection fraction spectrum [7,8,9,10].

Although the mechanisms responsible for the benefits of SGLT2 inhibition have only partly been elucidated and are still a matter of research, the results of many pivotal studies have inevitably affected cardiology practice and have been recognized in the most recent international guideline releases, with special focus on HF management [11]. Since the principal pharmacologic action of these drugs occurs by blocking SGLT 2 function in the glomerular tubule proximal segment, preventing the reabsorption of filtered glucose coupled with Na^+^ and thus decreasing the overall renal glucose reabsorption by approximately 60% and overall kidney Na^+^ reabsorption by less than 5% [1], the question that arises is, “Do those pharmacologic effects provide a reasonable explanation of the observed extension of clinical benefits?”. The question is an important one: SGLT2i is such a potent class of drug, that prescribing it entails responsibility in deciding its use as a therapy and demands we understand the manifold action this class of drug can provide in clinical conditions where kidney function is involved and corrupted to a varying extent.

## 2. Renal Physiology: A Unique Example Based on Efficient Control of Oxygen Consumption

It is commonly recognized that the heart and kidneys both share pathophysiological mechanisms that generate clinical conditions; however, little attention is paid to the evidence that their performance is designed to satisfy opposing demands. The heart has to provide each organ and apparatus with oxygen and nutrition on the basis of intercurrent demand without any predefined schedule, while the kidneys have to adapt the body fluid and electrolyte content to the physiological needs in order to cope with changes in the internal and external environment. The heart and kidney thus cope with two different physiological needs, namely the energy requirements of the organ versus the function balance of the organ. 

When one compares the renal and cardiac physiology, the similarities and differences become apparent, as the cardiac and renal mass are approximately the same (0.4–0.5% of body mass), as well as the metabolic rate (Ki), which is the highest in the body Ki 440 (in Kcal/Kg/day), while O_2_ consumption is higher in the heart at 11% versus 7% [12,13,14], although the blood flow is in favor of the kidneys at 25% versus 7% [14,15,16] (Table 1).

The peculiar heart structure, through the engagement of organ-specific mechanisms such as heart rate and the Frank–Starling law—prominent contributors [17] besides contractility and Laplace’s law—can swiftly increase the cardiac output upon demand via a large range of performances exploiting the strict relation to the phasic coronary flow, which may rise as much as 400% in trained subjects. The unmatched capacity of the coronary microvasculature to increase tissue delivery to the energy-craving cardiomyocyte metabolism is provided by physical training through increased arteriolar densities and/or diameters, together with the formation of new capillaries that maintain vasculature at a level commensurate with the degree of exercise-induced physiological myocardial hypertrophy. On top of this, physical training alters the distribution of coronary vascular resistance, so that more capillaries are recruited, resulting in an increase in the permeability-surface area product, regardless of the numerical capillary density [16].

Kidney physiology works differently and despite the much higher arterial blood flow, the unique renal structure and physiology combine to maintain the filtrate at around 180 L daily with an approximate glomerular filtration rate (GFR) of 125 mL/min, while the daily urine output may vary from 0.5 to 2.5 L/day in relation to the dryness of the ambience. It has been shown that renal O_2_ extraction remains stable over a wide range of renal blood flow [18], indicating that the high renal arterial flow undergoes strict regulation at different levels. The extensive renal sympathetic innervation largely governs the neurohumoral system response, which ensures kidney perfusion, affecting the circulation pressure [19], while peculiarities of the vascular architecture trigger preglomerular diffusional shunting of oxygen from arteries to veins, regulating the arterial flow directed toward the nephrons. Besides the above-mentioned regulatory systems in the kidneys, the glomerular hemodynamics and/or transport processes are balanced by the local release of various autacoids (nitric oxide, bradykinin, endothelin, angiotensin II, and prostanoids, to name a few) [20].

In the kidneys, the wealth of regulatory mechanisms can provide continuous adjustment of vasoconstriction and vasodilation in the afferent and efferent arterioles, counteracting wide-ranging changes in the blood pressure and keeping the intraglomerular pressure stable. These regulatory mechanisms are the main factors behind GFR generation, but the complexity of renal physiology regulation makes it largely impossible to predict or assess O_2_ consumption [21]. It is striking how the kidney labors to provide continuous checks and balances for body fluid and circulatory pressure. Indeed, the physiology of the kidneys is specifically designed to provide efficient blood clearing, while maintaining the whole circulatory balance without compromising O_2_ distribution to its own highly specialized tissues as they perform their endocrine, metabolic, and reabsorbing functions. 

These highly specific renal activities are sustained by adenosine triphosphate (ATP) production using largely aerobic mechanisms (roughly 95%), whereas some nephron segments, particularly in the less perfused medulla, can use anaerobic metabolism more efficiently [19,20,21]. 

We must remember that renal blood flow is the main contributor to determining the glomerular filtration rate (GFR), and as the plasma Na^+^ concentration is relatively constant in normal subjects, GFR directly determines Na^+^ load in the glomerular filtrate. Given that the kidney reabsorbs 99.5% of filtered Na^+^ via activation of the ATP-dependent Na^+^/K^+^ pump, the renal O_2_ consumption driven by Na^+^ reabsorption is proportionate to the GFR [21,22]. The following deserves consideration: if we assume a renal filtrate of 180 L/day, in the presence of 140 mmol of Na^+^ plasma concentration, the kidneys have to reabsorb ≈ 25 moles of Na^+^. As one mole of O_2_ can generate 3 moles of ATP and 1 mole of ATP is needed to reabsorb 9 moles of Na^+^, the kidneys need ≈3 moles of O_2_ (corresponding to 96 gr of O_2_) to reabsorb 99.5% of the Na^+^ content in 180 L of filtrate [23]. It has been estimated that the energy required to reabsorb 1 mole of Na^+^ against an electric potential of −70 mV in the cytoplasm and chemical gradient is about the same as that required to lift 1 mole of Na^+^ (≈23 g) to a height of approximately 70 km [21]. As Na^+^ reabsorption mainly occurs in the renal cortex and any increase in filtrate leads to an increase in Na^+^ load reabsorption, filtrate production is increased, O_2_ consumption increases mainly in the cortex, but may also affect O_2_ delivery to the medulla.

**Table 1 ijms-24-09957-t001:** Comparison between heart and kidney-specific biological and metabolic characteristics [5,12,13,14,16,17,21]. See the text for the details. * The coronary microvasculature can increase tissue delivery by physical training through increased arteriolar densities and/or diameters, together with the formation of new capillaries; it can alter the distribution of coronary flow, increasing the permeability surface area product. ** Based on: Starling law, Contractility, Laplace law and heart rate. GFR: glomerular filtration rate; OMR: organ metabolic rate; RBF: renal blood flow.

Organ	Total Blood Flow (L/min)	Parenchimal Flow at Rest (mL/min)	Parenchimal Flow Reserve (mL/min)	Proportion of Total Body O_2_ Consumption at Rest (%)	OMR Ki (kcal/Kg/day)	O_2_ Extraction (%)	Organ O_2_ Consumption(%) Na/ATPase Ca ATPase Other		
**Heart**	4/6	Coronaric 220/260	Coronaric 450/600	11	440	75	1–5	15–30	- Actinomyosin ATPase—40–50 - Proton leak-15
**Kidney**	1.2	Glomerular 90/120	Glomerular 100/140	6–7	440	10–15	70	_	Gluco- neogenesis
**Organ**	**Flow Regulation** **in Delivering O_2_** **to the Organ**			**Maximum** **increase in O_2_** **Consumption (%)**			**Prominent Regulator of Performance Maintenance**		
**Heart**	Phasic coronary flow: - Strictly dependent on the magnitude of the difference between arterial and tissue pressure. - This reflects the combined effects of cavity-induced extracellular pressure and shortening the induced intramyocyte pressure (tissue pressure), leading to vascular compression.			~ 400% * - The increased O_2_ demand ** is followed by increased perfusion and the delivery of O_2_ is strictly correlated to the increased coronary flow.			Filling Volume Change (Frank O–Starling E law) - Normal blood circulation under a wide range of workloads, depending on the preload reserve.		
**Kidney**	Inhomogeneous blood flow: - Renal cortex highly perfused as a result of having the highest rate of NA-K ATPase-dependent Na reabsorption. - Only 10 to 15% of blood perfusion is directed toward the medulla to preserve osmotic gradients and enhance urinary concentration by exploiting the countercurrent mechanism.			Difficult to assess: - O_2_ delivery is related to renal blood flow responsible for the GFR. - In normal subjects the plasma Na^+^ concentration is relatively constant and determines the Na load in the glomerular filtrate (around 125 mL/min/1.73 m^2^). - The reabsorption of 99.5% of filtered Na^+^ is dependent on cortical Na/K ATPase. - As the Na^+^ reabsorption is proportionate to the GFR, the filtrate amount is mainly responsible for O_2_ consumption. When the GFR exceeds 135 mL/min/1.73 m^2^, a condition of hyperfiltration occurs, entailing a steep rise in kidney O_2_ consumption.			Filtration Pressure Change (Starling E law) - The renal high precapillary pressure provides the hydrostatic pressure reserve to preserve the GFR over a wide range of renal hemodynamic changes. - The ratio between GFR (expressing the O_2_ consumption related to active transport) and RBF (expressing the O_2_ delivery) is the filtration fraction, which does not vary to any large extent in humans under physiological conditions, hovering at around 20%.		

## 3. The Source of Glomerular Filtration Pressure and the Generation of the Filtration Fraction

Among the factors involved in generating the glomerular filtrate, glomerular ultrafiltration pressure plays a leading role. The pressure of glomerular filtration results from the relative combination of the hydrostatic and oncotic pressure across the capillary wall; this provides the energy for water exchange between the plasma and the interstitial fluid. Despite its precapillary origin, it is roughly twice the rate (60 mm Hg) [24] of the precapillary pressure elsewhere, due to the energy delivery needed to generate and maintain glomerular filtration over a wide range of renal hemodynamic changes. 

As the filtration fraction (FF) is the renal functional index measured by the ratio between the GFR mL/min (renal O_2_ consumption index; nominator) and the renal blood flow mL/min (RBF, renal O_2_ delivery index; denominator) ×100, by expressing the percentage of filtrate formed in the Bowman space, it represents the main index of kidney function balance, which normally hovers around 20%. If FF represents the space relative to the renal blood flow and does not vary to any large extent in humans under physiological conditions [21], this implies a near-constant tissue O_2_ tension (PO2), as normally occurs in the tissue. It also explains why a relatively mild GFR increase beyond 135 mL/min/1.73 m^2^ is commonly considered as a hyperfiltration condition affecting glomerulus integrity [25,26]. Taking all of these factors together, if GFR generation is intimately tied to RBF, and 99.5% of glomerular filtrate together with the Na^+^ solute have to be reabsorbed through a high energy-consuming process, the way to restore the kidney from a negative energy balance cannot be based on increasing the blood flow to the organ. On the contrary, one needs to lower the filtration production, as this will lessen the number of sodium ions that must be transported according to the oxygen delivered. From the physical forces that generate glomerular filtration, it is apparent that the simplest way to lower the filtration production is to reduce the glomerular filtration pressure and thus decrease FF.

## 4. The Role of Adenosine

### 4.1. Adenosine Receptors

Adenosine (ADO) is commonly recognized as the precursor to adenosine triphosphate (ATP), but it also serves as an autacoid that binds to cell surface receptors in the kidneys, heart, brain, retina, and skeletal muscle, mediating different organ functions. The ADO autacoid action on the membrane cell receptor consists of matching cell energy consumption with the available O_2_ [19,20]. According to this relationship, the interstitial concentration of ADO rises when the energy balance in the cell tissue becomes negative, leading to adenosine A2 receptor (A2R) activation. A2R stimulation activates local vasodilation, adjusting the blood flow to meet demand. ADO action is based on a short half-life and diffusion, so it is a fast and efficient method of matching supply and demand over a short range within an organ [19,20,27]. Because of the intricacy of renal physiology, it is not surprising that ADO can play a more complex role including differential effects on the renal cortical and medullary vascular structures; it has a specific role in tubuloglomerular feedback (TGF), and is involved in governing renin secretion and transport processes in the tubular and duct collecting system (Figure 1). 

As glomerular filtration pressure is the main force involved in GFR generation, the main contributor to the kidney’s energy outlay, ADO can efficiently decrease the filtration pressure by combining its action on vasoconstrictor adenosine 1 receptor (A1R) in the afferent arteriole and on vasodilatory A2R in the efferent arteriole. In the individual nephron, the response to ADO stimulation is based on domain nuances driving the activity around each of the two receptors. The wide range of differentiated responses provided by the two adenosine receptors have significant conceptual implications for kidney physiology. The vasoconstrictive action exerted by ADO with its A1R activation ends by decreasing the glomerular flow and hence decreasing the renal O_2_ burden for Na^+^ reabsorption, but also implies an overall decline in blood flow to the glomerular structure, including the portion located in the medulla, which is supplied by a limited flow, meaning that O_2_ delivery to the medulla may only withstand a more limited degree of variation. The point is consistent with the observation that A1R-mediated afferent arteriolar constriction prevails in the surface nephrons, whereas deep cortical nephrons, which supply the blood flow to the renal medulla, can respond to ADO by A2R-mediated vasodilation [19,28]. The differentiated nuance of the glomerular adenosine receptor response provides insight into the sensitive balance that presides over renal nutrition and O_2_ delivery, particularly in the medullary tubule segments that are more vulnerable to hypoxia and play a crucial role in urine concentration. This aspect seems consistent with data stemming from genetic studies performed in a mouse model with AT II-induced CKD. In these studies, ADO stimulation of the red blood cell A2B receptor succeeds in enhancing AMPK activation, leading to a release of 2.3-biphosphoglycerate (2.3-BPG), a specific erythrocyte metabolite that, by promoting delivery of tissue oxygen, counteracts kidney hypoxia and CKD progression [29] (Figure 1).

**Figure 1 ijms-24-09957-f001:**
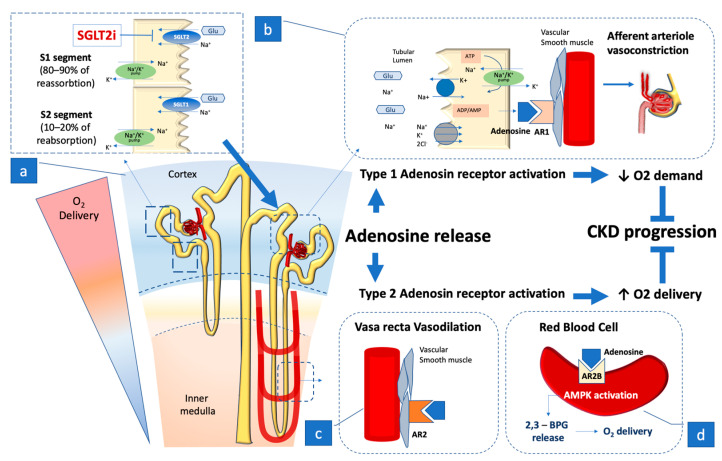
The largest O_2_ consumption (approximately 85%) [19,20,21] in the kidneys occurs in the cortex, where the reabsorption of Na^+^ and water mainly occurs in the proximal tubule (**a**). The figure summarizes the cascade of effects occurring with SGLT2 and SGLT1 engagement in sodium and glucose reabsorption in the early proximal tubule and in the renal medulla. In a normal glycemic environment, Na^+^ reabsorption linked to glucose reuptake corresponds to approximately 5% of overall Na^+^ reabsorbed in the kidneys. It can undergo fluctuating variation linked to timely changes in the plasma glucose concentration, and affect Na+ concentration in the filtrate, which is sensed by the juxtaglomerular apparatus through Na+ reabsorption via activation of the ATP-dependent Na^+^/K^+^ pump, leading to adenosine (ADO) release [19,20,21] (**b**). The amount of liberated ADO engages specialized adenosine 1 receptor (A1R), located in the afferent arteriole, and adenosine 2 receptor (A2R), placed in the efferent arteriole and in the vasa recta situated in the medulla. A1R in the afferent arteriole causes local vasoconstriction decreasing blood flow in the glomerulus and thus affecting the glomerular filtration pressure and the rate of filtrate production [19,20,21] (**c**). The A2R located in the efferent arteriole and in the vasa recta causes a vasodilating action in the vasculature with a bidirectional effect by vasodilating the efferent arteriole it contributes to lowering the glomerular filtration pressure, and by coupling vasodilation of the vasa recta it preserves the flow to the renal medulla where oxygen delivery is less copious. The simultaneous ADO action on both receptors constantly regulates the local O_2_ consumption through the juxtaglomerular feedback, avoiding imbalance in the metabolic needs of the kidneys [19,20,21] (**d**). Recently, a further action of ADO was observed on the AR2 receptor located on the red blood cell membrane where ADO stimulation enhances AMPK activation, leading to a release of 2.3-biphosphoglycerate (2.3-BPG), a specific erythrocyte metabolite that promotes delivery of oxygen to tissue, counteracting local kidney hypoxia and CKD progression [29].

### 4.2. Adenosine Action, Macula Densa, and Renin–Angiotensin–Aldosterone System Activation

While rapid minor shifts in blood volume and arteriolar tone are managed via the baroreceptor reflex, in the long term, regulation is based on the renin–angiotensin–aldosterone system (RAAS), which has a central action in the kidney and can alter blood volume and circulation balance chronically. Although the renin–angiotensin system is present in several organs including the peripheral vessels, the primary source of renin is the juxtaglomerular apparatus (JGA), packed specialized cells, commonly named macula densa, lining the wall of the distal tubule at the point of glomerulus contact (Figure 2). This cell pack contains the largest store of renin involving the kidney as the center regulating RAAS activation [30], as these specialized cells complement JGA intervention on single nephron function and structure. The macula densa senses the increase in luminal Na–Cl–K delivery via an NKCC2-based process, leading to an increase in the basolateral release of ATP, which in turn is converted by endonucleotidases CD73/39 to adenosine (ADO) [22]. The ADO generated activates A1R in the adjacent afferent arteriole, decreasing the vessel section and the glomerular filtration pressure. The ADO effect on glomerular filtration increases the distal delivery of sodium chloride, which also increases the hydrostatic pressure in the Bowman space, further attenuating the filtration pressure [19,21].

In the proximal tubule, ADO also stimulates the transport of Na^+^, decreasing the Na^+^ load to the tubule segments that extend into the less oxygenated medulla. In these tubule sections, ADO locally activates A1R, inhibiting Na^+^ transport up to the thick ascending limb of the medulla, and enhances the medullary blood flow via A2R activation, which increases local O_2_ delivery and limits O_2_-consuming Na^+^ transport. The ADO action in decreasing the glomerular filtration pressure exerts its effects not only by decreasing O_2_ consumption in the renal cortex through lowering the GFR, but also by partially inhibiting the function of other transporters such as the Na^+^- H^+^-exchanger NHE3 [19]. To accomplish the ordinary check and balance system of renal physiology, ADO activation of A1R also causes prolonged inhibition of renal renin secretion into the bloodstream [31], preventing angiotensin (AT) II with its vasoconstrictor action [32,33] (Figure 2). 

ADO’s inhibitory action on renin release has considerable weight in the overall circulatory balance, which transpires when dysregulation occurs with the onset of HF. When HF sets in, leading to decreased renal perfusion, the JGA cells react by swiftly spilling renin into the bloodstream, which activates the metabolic chain leading to AT II production. The principal effect of AT II in the kidney is to restrict the section of renal vasculature, its main action being on the lesser-sized efferent arteriole, leading to a hydraulic filtration pressure increase. In this setting, ATII outstrips the ADO action on the afferent arteriole and becomes the principal contributor to a filtration fraction increase, the leading mechanism involved in GFR maintenance as the HF syndrome evolves [34,35]. In the course of HF progression, the increasingly low renal blood flow increases GFR generation dependence through the action of AT II, which raises the glomerular filtration pressure gradient, curbing the plasma volume that is exposed to the pressure gradient at any given time per area unit of the capillary wall. The net effect is an increased plasma protein concentration, which raises the oncotic pressure at the level of the efferent arteriole and hence causes a rise in FF. The rise in FF will attenuate the absolute drop in the single nephron GFR, even without neurohumoral interference or ultrafiltration pressure change. However, as an ultrafiltration equilibrium is reached when the maximum FF is achieved at ∼60%, a further decrease in renal blood flow causes a linear GFR dip as the ultrafiltration pressure gradient cannot be maintained over the entire length of the glomerular capillary, which results in part of this capillary no longer being used for ultrafiltration and restricting the glomerular filtration area [36]. With HF in progress, AT II counteracts the inhibitory ADO effect on NHE3, leading to retention of Na^+^, which increases the osmolarity of the blood, and to a shift of fluid into the blood volume and extracellular space [37]. Other unfavorable actions linked to high ATII renal concentration are not only the increased production of superoxide anion due to angiotensin I receptor activation [38], which in turn triggers metabolic oxidative stress leading to loss of passive Na^+^ reabsorption, but also impairment of the paracellular permeability in the tubule [23] (Figure 2). 

It is worth noting that in the early phase of SGLT2 inhibitors administration, the osmotic diuresis coupled with increased natriuresis resulted in systemic plasma renin activity elevation [38], which was no longer present after six months of treatment [39] and was not coupled with a significant change in the aldosterone-to-renin ratio [39,40], excluding any detrimental neuro–hormonal activation. On the basis of current available data, SGLT2 inhibition seems able to transiently activate the systemic plasma rennin activity, without the concomitant intrarenal rennin activation [41].

With everything considered together, AT II actions resulted in increased kidney O_2_ consumption, leading to relative intrarenal hypoxia with counteracting activation of the hypoxia-inducible (HIF) system, which is based on two inducible key mediators in cellular oxygen homeostasis, HIF-1 and HIF-2. The two factors have functions that only partially overlap as the glycolytic genes are predominantly regulated by HIF-1, whereas HIF-2 is the main regulator of hypoxic vascular endothelial growth factor and EPO induction in tissues that express both HIF-1 and HIF-2. The HIF system is designed to provide a cytoprotective effect in acute ischemia-reperfusion injury, while in chronic renal hypoxia its response to inflammation takes a profibrotic role [42]. 

Consistently, in an experimental animal model based on knocking out SGLT2, the prolonged exposure of the medullary tract to increased Na^+^ concentration has been proven critical for the induction of renal growth and caused markers of renal injury, inflammation, and fibrosis to appear [43]. This condition seems expressed in type 2 diabetic patients who are hospitalized with acute kidney injury while they are receiving SGLT2 inhibitors. In these subjects, there have been reports of an increased finding of blood and urine levels of neutrophil gelatinase-associated lipocalin, originating from the distal tubular segments, whereas in the blood and urine, kidney injury molecule 1, a biomarker of proximal tubular damage, is unaltered [44]. The transport shift may, therefore, “simulate systemic hypoxia” and the lower oxygenation in the corticomedullary junction of the kidneys may trigger the hypoxia-driven hypoxia-inducible factor 2α in the interstitial cells, activating an observable increase in erythropoietin expression [45], which may improve oxygen delivery to the outer medulla and facilitate oxygenation of the heart and other organs.

As the emunctory function of the kidneys is to preside over the vital regulation of body fluid and electrolyte content, the renal effect of AT II reflects a nephrocentric reaction designed to preserve the GFR, providing an explanation for why it is immediately preserved in the low output state. The kidney reacts to a decrease in its own blood supply, similar to other organs, by releasing renin as a local factor, but it also unleashes an AT II renal and systemic circulatory response that bolsters the preservation of GFR [46]; unfortunately, this occurs at a soaring cost of O_2_ consumption, which impacts the strict kidney balance in terms of energy expenditure. In the end, it is not surprising that the kidneys’ consumption of O_2_ may independently steer the HF outcome.

Independently of glycemic status, the ADO generated by the JGA exerts a minute per minute control over the nephron filtration pressure and renin release. This peculiar control provides a reason for SGLT2 inhibition being so effective at maintaining renal O_2_ balance and for defusing the neurohormonal activation, sparing HF outcomes in diabetic and not diabetic populations [47].

## 5. SGLT2 Activity and Sympathetic Drive

### 5.1. Effects on Sympathetic Activity

In the clinical progression of HF, the renal sympathetic tone is greatly enhanced and the kidneys are the main contributor to the norepinephrine spillover, directly affecting the clinical outcome of HF [48]. The progression of CKD is also intensively marked by renal sympathetic traffic enhancement, partly related to the activation of α-2 adrenoreceptors and partly to the release of intrarenal AT II. Renally synthesized AT II regulates organ function in a paracrine fashion by modulating Na^+^ and water reabsorption within the proximal tubule together with systemic AT II [49]; consistently with this, renal sympathetic denervation abrogates the reabsorption of Na^+^ in the proximal tubule [50], where active natriuretic peptides are fast degraded by highly concentrated neprilysin, which is also responsible for the degradation of other peptide hormones such as angiotensin I and II, endothelin [51]. As well as direct kidney management of Na^+^ reabsorption, the increased sympathetic traffic in the proximal tubule augments the expression of both NHE3 and SGLT2, playing a further role in the avid sodium reabsorption involved in HF progression [52]. SGLT2 inhibitors interfere with both SGLT2 and NHE3 in the proximal tubule [28], yielding short-term increases in the fractional excretion of sodium [53]. On the plus side, SGLT2 inhibition attenuates renal sympathetic activity and reduces the renal norepinephrine content in states of experimental nutrient excess [28,54], while in animals with HF, renal denervation attenuates the magnitude of response to SGLT2 inhibition [52]. These observations place SGLT2 inhibitors as functional antagonists to renal sympathetic nerve hyperactivity in HF, which is the main contributor to norepinephrine spillover [55] and is closely linked to outcome in HF patients [56].

### 5.2. SGLT2 and SGLT1 Synergy and “Off Target” Implications

In renal physiology, glucose reabsorption is based on a highly efficient architecture placing SGLT2 in the early proximal tubule to perform the bulk of glucose reabsorption (~80–90%) and positioning sodium glucose cotransporter 1 (SGLT1) in the late proximal tubule, where it reabsorbs the amount of glucose that escapes SGLT2. In the final section of the tubule, the fluid in the glucose concentration falls below the line of stoichiometric reabsorption of 1 glucose molecule to 1 Na^+^ performed by SGLT2. As Na^+^–glucose cotransport is electrogenic, in the later section of the tubule, glucose reabsorption requires SGLT1 to increase its sugar concentration strength, exploiting the electrical power provided by 2 Na^+^ ion, so as to transfer 1 glucose molecule from the filtrate into the bloodstream, thereby doubling the energy expense of reabsorption. 

As the normal daily glomerular filtrate contains ~1 mol of glucose (~180 g) and the combined action of cotransporters has the capacity to reabsorb ~2.5 mol glucose per day (~450 gr per day), this suggests that in nature, the function of SGLT2 has been dimensioned to cater to broad variations in the glucose concentration that can greatly affect Na^+^ reabsorption, with implications for renal O_2_ consumption [19]. Persistentent augmenting of Na^+^ glucose reabsorption leads to proximal tubule hypertrophy, a primary cause and effect of glomerular hypertrophy driven by hyperfiltration [1,21,26].

The peculiar extensibility of SGLT2 and SGLT1 action has important “off-target” effects that stem from their effect on proximal tubular Na^+^ transport, where bicarbonate accounts for approximately 80% of Na^+^ reabsorption in this portion of the nephron. The rise in sodium–glucose reabsorption may affect body fluid retention such as sodium bicarbonate, driving additional passive sodium chloride reabsorption. Note that in diabetics, the effect of SGLT2 pharmacologic inhibition is partially offset by enhanced SGLT1 activation, while investigation in animals has provided conclusive evidence that SGLT1 can reabsorb ~30% of filtered glucose, explaining why SGLT2 inhibitors never produce the amount of glucosuria expected if SGLT2 were completely inhibited (preventing 80–90% reabsorption of the filtered glucose load) [57]. This point focuses on the peculiar role SGLT1 plays in the kidney, as SGLT1 engagement can prevent or just limit glucose loss in the urine, at the cost of high energy expenditure by the kidneys.

Intriguingly, two studies performed in Akita mice support the role of SGLT1, expressed in the membrane of the tubuloglomerular apparatus, in increasing nitric oxide (NO) S1-dependent NO formation by sensing the glucose concentration reaching the macula densa, thereafter reducing the vasoconstrictor tone set by TGF and contributing to glomerular hyperfiltration [58]. In Akita mice, the absence of SGLT1 in the tubuloglomerular apparatus not only lowers glomerular hyperfiltration, but also reduces kidney weight, glomerular size, and albuminuria [59]. These findings suggest that SGLT1 may have implications for renal structure and performance besides the reabsorption of glucose.

The inhibition of SGLT2 can reduce glycemia and insulin resistance, and can lower the availability of cellular glucose, while regardless of basal hyperglycemia, it can stimulate a starvation-like response. The response includes SIRT1/AMPK (Sirtuin1/adenosine monophosphate-activated protein kinase) activation and inhibition of the protein kinase b/mTOR1 (mammalian Target Of Rapamycin) pathway [60]. This specific activation, in inducing autophagy, promotes cellular defense and pro-survival mechanisms that counteract the primary pathophysiological mechanism of proximal tubule hypertrophy in diabetes and in conditions involving insulin resistance [20,26,28]. In experimental models and in patients with T2DM, urine metabolomics have indicated that the inhibition of SGLT2 induces a metabolic shift from glycolysis to more mitochondrial oxidation [20,22].

Studies in non-diabetic mice suggest that the kidney’s metabolic response to SGLT2 inhibition compensates (a) for the partial inhibition of tubular NHE3 and the glucose uptake/urinary glucose loss, including renal gluconeogenesis upregulation, and (b) urinary Na^+^ loss, by inducing tubular secretion of the tricarboxylic acid cycle intermediate, alpha-ketoglutarate, which communicates to the distal nephron the need for compensatory Na^+^ reabsorption [61].

SGLT2 inhibition shifts some of the glucose, Na^+^, and fluid reabsorption downstream, providing a more equal distribution of transport work and mimicking systemic hypoxia to the renal oxygen sensor, triggering the upregulation of renal NHE3. This effect in HF and on the remaining nephrons in CKD could enhance the natriuretic efficacy and renal hemodynamic effect of SGLT2 inhibition and thereby contribute to kidney and cardio protection in nondiabetic patients. 

The inhibition of SGLT2 lowers body weight by coupling the initial natriuretic effect with renal glucose loss, which shifts substrate utilization from carbohydrates to lipids and reduces body fat, lessening visceral and subcutaneous adiposity [62]. This effect also augments the release of free fatty acids, leading to ketone body formation, which can be used as an additional more efficient energy substrate both in the kidneys and failing heart [63]. At the same time, the transport shift to the straight proximal tubule and thick ascending limb in the renal outer medulla could reduce the O_2_ availability, endangering medullary structures, as mentioned above [43,44,45].

As gliflozins are the only class of hypoglycemic drugs combining glycosuric and natriuretic actions, they play a joint role in vascular fluid restriction and hemoconcentration. By inhibiting glucose renal reuptake, this class of drug induces significant osmotic diuresis, which selectively decreases the volume of the interstitial space between cells (known as the third space) and affects body weight beyond what nutrient loss can provide [1,20,28]. Osmotic diuresis obtained with SGLT2i has been proven in humans through a double-blind randomized study conducted on 59 type 2 diabetics [62]. It has been postulated that SGLT2i may regulate both the volume between the interstitial space and the vascular bed (interstitial > intravascular) in HF, thus reducing the neurohumoral stimulation generated by the signal denoting decreased vascular filling activated by the baroreceptors [62]. Note that, unlike loop diuretics that promote natriuresis by inhibiting carbonic anhydrase in the thick ascending limb, SGLT2 inhibition halts Na^+^ and glucose reabsorption driven by the activation of the Na^+^/K^+^ pump in the brush border, curbing the O_2_ consumption rate in the critical cortex area, without any impact on the electrolyte balance. The peculiarity and sequence of pharmacologic mechanisms activated by SGLT2 inhibitors and by loop diuretics suggests a reason their combined action generates reciprocal potentiation of natriuresis [63] and supports their combined use in acute decompensated HF.

### 5.3. Other Implications of SGLT2 Inhibition and RAAS Interaction

One should note that a significant increase in kidney sensitivity to adenosine-induced vasoconstrictive action is caused by inhibiting the synthesis of local vasodilatory molecules such as NO or prostaglandins [19,20,27], so that non-steroidal anti-inflammatory drugs (NSAIDs) are among the substances that can lead to acute kidney injury through the potentiation of ADO action on A1R. 

The inhibition of SGLT2 not only decreases renal cortex O_2_ consumption, as a consequence of lowering GFR, but through partial functional inhibition of other transporters such as the Na- H- exchanger NHE3 [21,28], in the brush border. The co-inhibition of NHE3 contributes to augmenting natriuresis and to lessening blood pressure, with lower effect of SGLT2 inhibitors in the non-diabetic setting [20,64]. It has to be noted that SGLT2 inhibitors enhance renin levels and vasopressin (or copeptin) levels and reduce renal free-water clearance in animal models and humans [65,66,67,68]. This effect is associated with the increased renal protein expression of vasopressin V2 receptors and phosphorylated aquaporin-2 in rats [66,67], indicating active compensation to counter the diuretic and natriuretic effects and highlighting the intricate check and balance system of the renal emunctory function.

At the same time, the inhibition of SGLT2 also significantly increases urate excretion, bringing about an indirect effect on the urate transporter URAT1 in the proximal tubule brush border [28,69].

## 6. SGLT2 Activity and Acute Kidney Injury

There have been some concerns about the possible association between SGLT2i and an increased risk of acute kidney injury (AKI). This has been mainly the consequence of the reports of the US Food and Drug Administration Adverse Event Report System (FDAERS) [70,71]. A possible explanation for these events is related to osmotic diuresis, which increases the risk of hyperosmolarity and dehydration. A second hypothesis could be represented by the absorption of the increased tubular glucose by the glucose transporter GLUT 9b, which is present at the level of the proximal tubular cells, in exchange for uric acid. The consequent increased uricosuria could favor AKI through both crystal-dependent and crystal-independent mechanisms, particularly in some clinical conditions such as the use of radiocontrast, rhabdomyolysis, heat stress, and dehydration [71]. Another hypothesis is related to the high glucose concentration expression induced by aldose reductase, which is osmolar sensitive. This induction can, in turn, lead to the generation of sorbitol and fructose, which can be metabolized by fructokinase, leading to the synthesis of uric acid, oxidative stress, the release of chemokines, and local tubular injury and inflammation [72]. Finally, sorbitol and fructose can also cause the depletion of intracellular organic osmolytes, such as myo-inositol and taurine, which could contribute to the occurrence of AKI [73].

Despite FDAERS reports and the possible pathophysiological background, data generated by a randomized, double-blind, placebo-controlled crossover study, performed with magnetic resonance in type 1 diabetic patients, are reassuring. The study was designed to assess the acute effects on kidney tissue oxygenation and perfusion of a single 50 mg dose of dapagliflozin, and displayed improved renal cortical oxygenation without changes in renal perfusion or blood flow. This suggests the improved renal cortical oxygenation was linked to reduced tubular transport workload in the proximal tubules [74]. The improved O_2_ consumption in the kidneys may explain the long-term beneficial renal effects seen with SGLT2 inhibitors both in randomized trials and in observational studies, where the risk of AKI was reduced rather than increased after SGLT2 inhibition. Indeed, in a meta-analysis using data from the EMPA-REG OUTCOME, CANVAS, DECLARE TIMI 58, and CREDENCE trials, AKI risk was reduced by 25% [75]. Analogously, a reduced risk of AKI was observed in “real-world analyses” comparing SGLT2 inhibitors with the other hypoglycemic drugs [76,77].

## 7. Conclusions

We recently found evidence that the renal oxygen demand is critically linked to the circulation neuro–hormonal balance, and conditions that increase glomerular filtration pressure affect renal oxygen balance, leading to progressive glomerular damage and loss of filtration power, which primarily affect the cardiovascular outcome. In kidney physiology, the reabsorption of filtered glucose is primarily taken up into peritubular capillaries and returned to the systemic circulation or supplied as an energy source to further distal tubular segments. Recent studies have provided insights regarding the coordination of renal glucose reabsorption, formation, and usage. Moreover, a better understanding of renal glucose transport in disease states is shedding light on the complex interaction between the mechanisms of sodium reabsorption and renal oxygen demand [1,19,20,21]. The kidneys’ capacity to provide complete glucose reabsorption is a remnant of the ancient evolutionary tendency to avoid glucose loss, which was a scanty energy resource until the industrial revolution. 

A high daily intake of glucose leads to maladaptive kidney response in an effort to reuptake the excess of filtered glucose. This effort reveals the pathophysiological potential of SGLT2, which links the retention of glucose to the reabsorption of sodium, affecting the volume status and impairing TGF by increasing the glomerular flow through the vasodilation of the afferent arteriole and unbalancing the kidney oxygen demand. 

As the body’s evolutionary ability adapted to survival in environments with scanty energy resources, extra-calorie loss following the inhibition of renal glucose reabsorption inducing sugar spill into the urine can activate a mechanism resembling fasting. This action not only results in lowering glycemia, but also reduces the exceedingly high renal O_2_ consumption that is the prominent condition affecting the renal biology. 

While this concept contributes to the unexpected rationale of SGLT2 inhibition in the diabetic kidney, the action of SGLT2 can efficiently affect renal O_2_ consumption whenever cardiac or intrinsic kidney disorders result in a need to increase the glomerular filtration pressure in order to maintain the filtrate. 

The benefits observed after the administration of SGLT2 inhibitors in diabetics and non-diabetics with and without HF and or CKD [2,3,4,5,6,7,8,9,10] are probably the culmination of the primary metabolic/renal and secondary myocardial benefits as described above. Among them, rebalancing of the renin angiotensin system and the stimulation of erythropoietin may contribute to our understanding of the kidneys’ need to maintain its function and preserve the physiological balance of the organism.

Although many questions remain unanswered, such as the systemic metabolic consequences of losing glucose into the urine and the metabolic responses of the early proximal tubule and the down-stream segments, a new avenue in the treatment of cardiovascular and metabolic disorders has now opened and deserves close attention.

To remain within the metaphor of glucose being a friend or foe, the answer may simply be that over the course of evolution, glucose became more than a friend to mammals, a condition that entails certain risks, as discussed in our paper.

## Figures and Tables

**Figure 2 ijms-24-09957-f002:**
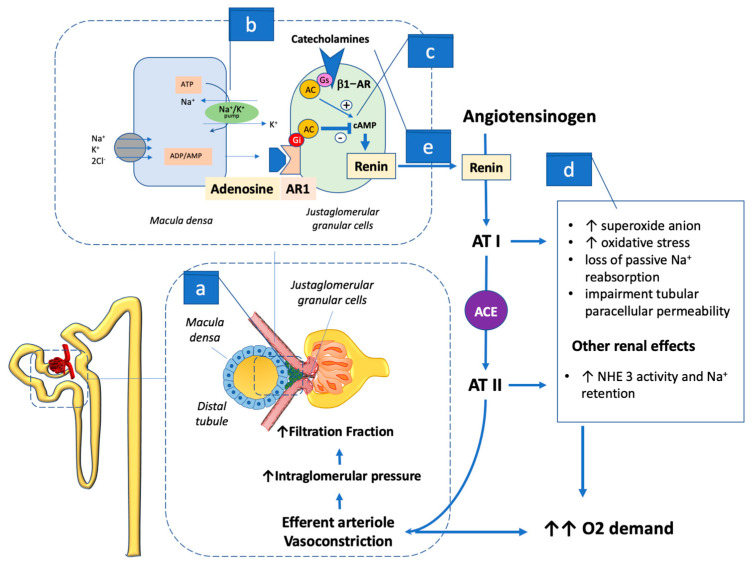
The figure shows how ADO constantly regulates glomerular filtration in each nephron. In the presence of euglycemia, the Na^+^ amount (**a**) in the filtrate reaches the juxtaglomerular apparatus and transits across the macula densa cells. (**b**) These cells sense the Na^+^ concentration by re-uptaking the electrolyte via the activation of the Na^+^/K^+^ energy based by breaking down adenosine triphosphate (ATP) to adenosine (ADO). The ADO that is generated binds the adjacent adenosine type 1 receptor (AR1) in the afferent arteriole leading to vessel section restriction. This action balances the intraglomerular pressure and thus the filtration fraction, which normally does not vary to a large extent from 20% under physiological conditions (see Section 3 in the text for details). (**c**) In the juxtaglomerular apparatus, ADO AR1 stimulation through inhibitory G (Gi) protein action mediates the inhibition of renin secretion via cAMP-dependent adenylate cyclase (AC) inhibition. This effect not only prevents systemic activation of the renin angiotensin (AT) aldosterone axis, but also affects efferent arteriole vasoconstriction maintaining stable filtration pressure and (**d**) limits the detrimental action of AT I and AT II on cell metabolism and on Na^+^ retention (see text for details). Of note, (**e**) AT II stimulates strong catecholamines release that in turn, via stimulatory G (Gs), activates cAMP, leading to AC activation and renin release by the juxtaglomerular apparatus, and in turn further enhancing the neuro–hormonal response. ACE: Angiotensin converter enzyme; ADO: adenosine; AR1: type 1 adenosine receptor; AT: angiotensin; Gi: inhibitory protein G; Gs: stimulatory protein G; ATP: adenosine triphosphate.

## Data Availability

Not applicable.

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
