# Peer review of "Renal Oxygen Demand and Nephron Function: Is Glucose a Friend or Foe?"

_ijms, 2023, doi:10.3390/ijms24129957_

Round 1

Reviewer 1 Report

Authors tried to summarise Renal oxygen demand and nephron function affected by glucose. In the renal physiology view, the work is meaningful. However, the link among each part of the review were not clear enough. The kidney perspective on the mechanism of action of sodium glucose co-transporter 2 inhibitors have been clarified by previous studies, including tubuloglomerular feedback, Blood pressure effects,Workload and hypoxia,Nutrient deprivation and ketogenesis, and Other protective mechanisms. The innovation point of this review should be the Renal oxygen regulators, so the introduction part should avoid overemphasizing the role of SGLT2i. Besides, in the SGLTi part, authors should show a overall schmatic mechasim figure of all regulators on Renal oxygen.

none.

Author Response

Authors tried to summarise Renal oxygen demand and nephron function affected by glucose. In the renal physiology view, the work is meaningful.

Response:

We would like to thank the reviewer for his/her comments. This is our point-to-point reply:

However, the link among each part of the review were not clear enough. The kidney perspective on the mechanism of action of sodium glucose co-transporter 2 inhibitors have been clarified by previous studies, including tubuloglomerular feedback, Blood pressure effects, Workload and hypoxia,Nutrient deprivation and ketogenesis, and Other protective mechanisms. The innovation point of this review should be the Renal oxygen regulators, so the introduction part should avoid overemphasizing the role of SGLT2i. Besides, in the SGLTi part, authors should show a overall schmatic mechasim figure of all regulators on Renal oxygen.

Response:

In the introduction we tried to reduce the overemphasis about the role of SGLT2i by removing the sentence: “Note that gliflozines have displayed high efficacy in decreasing HF onset and exacer-bation following optimal therapy, results found within weeks of drug administration. The benefits of these molecules have also proved to affect the overall cardiovascular outcome in subjects with atherosclerosis as their most frequent background pathology, displaying a time-related association with the restraining of CKD progression.”

As far as the schematic figure about renal oxygen it should be noted that Figure 1 and 2 already try to summarize these mechanisms. 

We hope that also after changes suggested by the other reviewers, the paper could be suitable for publication.

Reviewer 2 Report

1. Provide suitable title for table 1.

2. SGLT2 inhibitors reduced the risk of acute kidney injury and the risk of cardiovascular death or hospitalisation for heart failure with similar effects in those with and without diabetes (Lancet. 2022 Nov 19;400(10365):1788-1801). However, no renal protection benefits in non-DM patients are noted with other hypoglycemic therapy. It is suggested that renal protection effects of SGLT2 inhibitors might not be related with lowing blood sugar.

3. This review was entitled with "Renal oxygen demand and nephron function: is glucose a friend or foe?". It is suggested to make a clear conclusion to answer "is glucose a friend or foe".

Author Response

We would like to thank the reviewer for his/her comments. This is our point-to-point reply:

  1. Provide suitable title for table I

Response:

We added the following title to Table I:

 “Comparison between heart and kidney specific biological and metabolic characteristics”

  1. SGLT2 inhibitors reduced the risk of acute kidney injury and the risk of cardiovascular death or hospitalisation for heart failure with similar effects in those with and without diabetes (Lancet. 2022 Nov 19;400(10365):1788-1801). However, no renal protection benefits in non-DM patients are noted with other hypoglycemic therapy. It is suggested that renal protection effects of SGLT2 inhibitors might not be related with lowing blood sugar.
    Response:

In response to the well taken reviewer’s point, we added to section 4.2 the following conclusive phrase and the reference,

“Independently by the glycemic status, the ADO generated by the JGA exerts a minute per minute control over the nephron filtration pressure and the renin release. This peculiar control provides the reason why SGLT2 inhibition is so effective in maintaining renal O2 balance and in defusing the neurohormonal activation, sparing HF outcomes in diabetic and not diabetic populations [39].

and we added the following reference:

  • Nuffield Department of Population Health Renal Studies Group; SGLT2 inhibitor Meta-Analysis Cardio-Renal Trialists' Consortium. Impact of diabetes on the effects of sodium glucose co-transporter-2 inhibitors on kidney outcomes: collaborative meta-analysis of large placebo-controlled trials. Lancet 2022, 400, 1788-1801.

  1. This review was entitled with "Renal oxygen demand and nephron function: is glucose a friend or foe?". It is suggested to make a clear conclusion to answer "is glucose a friend or foe".

Response:

By accepting the reviewer suggested point we added to “Conclusion” ending, the following:

“To remain within the metaphor of glucose being a friend or foe, the answer may simply be that over the course of evolution glucose became more than a friend to mammals, a condition that entails certain risks as discussed in our paper.”

Reviewer 3 Report

This is a nice review with adequate experimental data. I have the following comment to the authors:

1.                     The authors could add experimental and clinical-trial data to elucidate the effect of SGLT-2 inhibitors on the risk of acute kidney injury.

Author Response

We would like to thank the reviewer for his/her comments. This is our point-to-point reply:

This is a nice review with adequate experimental data.

Response:

Thank you for having appreciated our work.

  1. The authors could add experimental and clinical-trial data to elucidate the effect of SGLT-2 inhibitors on the risk of acute kidney injury.

Response:

Thank you for the very useful comment. We know that reports of AKI in the FDAERS have raised the clinical issue about this problem. However, in clinical trials and observational cohort studies, AKI risk was reduced rather than increased in SGLT2 inhibitor users. We added the following section at page 12.

6. SGLT2 activity and Acute Kidney Injury

There have been some concerns about the possible association between SGLT2i and an increased risk of acute kidney injury (AKI). This has been mainly the consequence of the reports of the US Food and Drug Administration Adverse Event Report System (FDAERS) [73,74]. A possible explanation of these events has been related to the osmotic diuresis, which increases the risk of hyperosmolarity and dehydration. A second hypothesis could be represented by the absorption of the increased tubular glucose by the glucose transporter GLUT 9b, which are present at the level of proximal tubular cell, in exchange for uric acid. The consequent increased uricosuria could favor AKI through both crystal-dependent and crystal-independent mechanisms, particularly in some clinical conditions such as use of radiocontrast, rhabdomyolysis, heat stress and dehydration [75]. Another hypothesis is that related to the high glucose concentration induced expression of aldose reductase, which is osmolar sensitive. This induction can in turn leads to the generation of sorbitol and fructose, which can be metabolized by fructokinase, leading to the synthesis of uric acid, oxidative stress, the release of chemokines and local tubular injury and inflammation [76]. Finally, sorbitol and fructose can also cause the depletion of intracellular organic osmolytes, such as myo-inositol and taurine, which could contribute to AKI occurrence [77].

Despite the FDAERS reports and the possible pathophysiological background, data generated by a randomized, double-blind, placebo-controlled crossover study, performed with magnetic resonance in type 1 diabetic patients, are reassuring. The study designed to assess the acute effects on kidney tissue oxygenation and perfusion of a single 50 mg dose of dapagliflozin, displayed improved renal cortical oxygenation without changes in renal perfusion or blood flow. This suggests the improved renal cortical oxygenation was linked to reduced tubular transport workload in the proximal tubules [78]. The improved O2 consumption in the kidney may explain the long-term beneficial renal effects seen with SGLT2 inhibitors both in randomized trials and in observational studies where the risk of AKI was reduced rather than increased after SGLT2 inhibition. Indeed, in a meta-analysis using data from the EMPA-REG OUTCOME, CANVAS, DECLARE TIMI 58, and CREDENCE trials, the AKI risk was reduced by 25% [79]. Analogously, a reduced risk of AKI was observed in “real-world analyses” comparing SGLT2 inhibitors with the other hypoglycemic drugs [80-81].

We also added the following references:

73. Hahn, K.; Ejaz, A.A.; Kanbay, M.; Lanaspa, M.A.; Johnson, R.J. Acute kidney injury from SGLT2 inhibitors: potential mechanisms. Nat Rev Nephrol 2016, 12, 711-712.

74. Sridhar, V.S.; Tuttle, K.R.; Cherney, D.Z.I. We Can Finally Stop Worrying About SGLT2 Inhibitors and Acute Kidney Injury. Am J Kidney Dis 2020, 76, 454-456.

75. Hahn, K.; Kanbay, M., Lanaspa, M. A., Johnson, R. J. & Ejaz, A.A. Serum uric acid and acute kidney injury: a mini review. J Adv Res 2017, 8, 529-536.

76. Bjornstad, P.; Lanaspa, M.A.; Ishimoto, T. Fructose and uric acid in diabetic nephropathy. Diabetologia 2015, 58, 1993-2002.

77. Kitamura, H.; Yamauchi, A.; Sugiura, T. Inhibition of myo-inositol transport causes acute renal failure with selective medullary injury in the rat. Kidney Int 1998, 53, 146-153.

78. Laursen, J.C.; Søndergaard-Heinrich, N.; de Melo, J.M.L. et al. Acute effects of dapagliflozin on renal oxygenation and perfusion in type 1 diabetes with albuminuria: A randomised, double-blind, placebo-controlled crossover trial. 2021, 37, 100895.

79. Neuen, B.L.; Young, T.; Heerspink, H.J.L., et al. SGLT2 inhibitors for the prevention of kidney failure in patients with type 2 diabetes: a systematic review and meta-analysis. Lancet Diabetes Endocrinol 2019, 7, 845-854

80. Cahn, A.; Melzer-Cohen, C.; Pollack, R.; Chodick, G.; Shalev, V. Acute renal outcomes with sodium-glucose co-transporter-2 inhibitors: real-world data analysis. Diabetes Obes Metab 2019, 21, 340-348.

81. Iskander, C.; Cherney, D.Z.I.; Clemens, K.K.; et al. Use of sodium-glucose cotransporter-2 inhibitors and risk of acute kidney injury in older adults with diabetes: a population-based cohort study. CMAJ 2020, 192, E351-E360.

Reviewer 4 Report

The authors Gronda et al. have presented a well written review on the impact of glucose on renal oxygen demand and its subsequent impact on nephron function. While a major focus of this review is centered on evidence from use of SGLT2 inhibitors and mechanisms that are impacted by their use, I feel that readers would benefit greatly if the authors could include more background on what happens to renal oxygen demand in disease states like diabetes. The authors have provided excellent background on how the kidney physiology efficiently controls oxygen consumption including evidence of dysregulation of the same resulting in aberrant cortical oxygenation/ oxygen consumption could bolster the section and provide better understanding when you get to the next sections.

The authors have also done a fairly good job in discussing the counter effects of SGLT2i use in type 2 diabetics that can result in AKI. However, I feel also talking about the benefits that it provides with regards to O2 consumption needs to addressed. Including information from the randomized, double-blind, placebo-controlled crossover study assessing the acute effects of a high dose of dapagliflozin on kidney tissue oxygenation and perfusion in patients with type 1 diabetes carried out by Jens Christian Laursen and colleagues [PMCID: PMC8343250] could be a valuable source to add to this review.

In section 4, where the authors discuss the role of adenosine and the RAAS regulation of oxygen consumption, the section almost feels incomplete and a separate entity from the rest of the paper. Since the title and introduction suggest that the authors want to focus on the role of glucose on the regulation of renal O2 demand/supply, I feel including evidence showing the link between hyperglycemia or more importantly SGLT2i use and change in Adenosine and RAAS regulation could benefit the readers. For eg. Mori and Ishizuka (https://doi.org/10.2337/db18-1196-P) and Schork et al [PMCID: PMC6451223] reported that SGLT2i acutely effect plasma renin activity. Since increased renin leads to increased oxygen demand could this explain some of the side effects we observe in type 2 diabetics. On the other hand Sawamura et al [PMCID: PMC7706199] showed that SGLT2 inhibitor administration yielded minimal effects on Plasma aldosterone-to-renin ratio. Adding this information along with other evidences can better tie in section 4 with the rest of the manuscript.

Author Response

The authors Gronda et al. have presented a well written review on the impact of glucose on renal oxygen demand and its subsequent impact on nephron function. While a major focus of this review is centered on evidence from use of SGLT2 inhibitors and mechanisms that are impacted by their use, I feel that readers would benefit greatly if the authors could include more background on what happens to renal oxygen demand in disease states like diabetes. The authors have provided excellent background on how the kidney physiology efficiently controls oxygen consumption including evidence of dysregulation of the same resulting in aberrant cortical oxygenation/ oxygen consumption could bolster the section and provide better understanding when you get to the next sections.

Response:

Thank you for having appreciated our work and for his/her suggestions.

  1. The authors have also done a fairly good job in discussing the counter effects of SGLT2i use in type 2 diabetics that can result in AKI. However, I feel also talking about the benefits that it provides with regards to O2 consumption needs to addressed. Including information from the randomized, double-blind, placebo-controlled crossover study assessing the acute effects of a high dose of dapagliflozin on kidney tissue oxygenation and perfusion in patients with type 1 diabetes carried out by Jens Christian Laursen and colleagues [PMCID: PMC8343250] could be a valuable source to add to this review.

Response:

We mentioned the study of Laursen in the manuscript section focused on AKI (see the new section 6 at page 12:

Despite the FDAERS reports and the possible pathophysiological background, data generated by a randomized, double-blind, placebo-controlled crossover study, performed with magnetic resonance in type 1 diabetic patients, are reassuring. The study designed to assess the acute effects on kidney tissue oxygenation and perfusion of a single 50 mg dose of dapagliflozin, displayed improved renal cortical oxygenation without changes in renal perfusion or blood flow. This suggests the improved renal cortical oxygenation was linked to reduced tubular transport workload in the proximal tubules [78].

We also added the following references:

78. Laursen, J.C.; Søndergaard-Heinrich, N.; de Melo, J.M.L. et al. Acute effects of dapagliflozin on renal oxygenation and perfusion in type 1 diabetes with albuminuria: A randomised, double-blind, placebo-controlled crossover trial. 2021, 37, 100895.

  1. In section 4, where the authors discuss the role of adenosine and the RAAS regulation of oxygen consumption, the section almost feels incomplete and a separate entity from the rest of the paper. Since the title and introduction suggest that the authors want to focus on the role of glucose on the regulation of renal O2 demand/supply, I feel including evidence showing the link between hyperglycemia or more importantly SGLT2i use and change in Adenosine and RAAS regulation could benefit the readers. For eg. Mori and Ishizuka (https://doi.org/10.2337/db18-1196-P) and Schork et al [PMCID: PMC6451223] reported that SGLT2i acutely effect plasma renin activity. Since increased renin leads to increased oxygen demand could this explain some of the side effects we observe in type 2 diabetics. On the other hand Sawamura et al [PMCID: PMC7706199] showed that SGLT2 inhibitor administration yielded minimal effects on Plasma aldosterone-to-renin ratio. Adding this information along with other evidences can better tie in section 4 with the rest of the manuscript.

Response:

We thank the reviewer for the suggestion and we add to section 4 the following :

“It is worth noting to recall in the early phase of SGLT2 inhibitors administration, the osmotic diuresis coupled with the increased natriuresis resulted in systemic plasma rennin activity elevation [38] that is not more present after six months treatment [39] and it is not coupled with significant change in aldosterone-to-renin ratio [39-40], excluding any detrimental neuro-hormonal activation. On the basis of current available data, the SGLT2 inhibition seems able to transiently activate the systemic plasma rennin activity, without the concomitant intrarenal rennin activation [41].”

We also added the following references:

39. Tanaka, H.; Takano, K.; Iijima, H.; Kubo, H.; et al. Factors Affecting Canagliflozin-Induced Transient Urine Volume Increase in Patients with Type 2 Diabetes Mellitus. Adv Ther 2017, 34, 436-451

39. Sawamura, T.; Karashima, S.; Nagase, S.; et al. Effect of sodium-glucose cotransporter-2 inhibitors on aldosterone-to-renin ratio in diabetic patients with hypertension: a retrospective observational study. BMC Endocr Disord. 2020, 20, 177.

40. Tanaka, H.; Takano, K.; Iijima, H.; et al. Factors Affecting Canagliflozin-Induced Transient Urine Volume Increase in Patients with Type 2 Diabetes Mellitus. Adv Ther 2017, 34, 436-451.

41. Ansary, T.M., Nakano, D.; Nishiyama, A. Diuretic Effects of Sodium Glucose Cotransporter 2 Inhibitors and Their Influence on the Renin-Angiotensin System. Int J Mol Sci 2019, 20, 629.

Round 2

Reviewer 1 Report

The authors' reply did not solve my comments .  This review just add the role of SGLT2i on already known Renal oxygen demand and nephron function. However, the both parts have been well summarized in previous studies. In my opinion, this review still could not be accepted in current form. Authors should reorganize the language to better show the topic of this review.

none.

Author Response

We are sorry for reviewer's comments. We tried to discuss at the best  the topic.

Reviewer 4 Report

The authors have addressed all concerns raised.

Author Response

Thank you for the comments.